# Examining Changes in Internal Representations of Continual Learning Models Through Tensor Decomposition

**Nishant Suresh Aswani**[*,†]    **Amira Guesmi**[†]    **Muhammad Abdullah Hanif**[†]    **Muhammad Shafique**[*,†]

[*]Department of Computer Science and Engineering, New York University Tandon, Brooklyn, USA
[†]eBrain Lab, Division of Engineering, New York University Abu Dhabi, Abu Dhabi, UAE

## Abstract

Continual learning (CL) has spurred the development of several methods aimed at consolidating previous knowledge across sequential learning. Yet, the evaluations of these methods have primarily focused on the final output, such as changes in the accuracy of predicted classes, overlooking the issue of representational forgetting within the model. In this paper, we propose a novel representation-based evaluation framework for CL models. This approach involves gathering internal representations from throughout the continual learning process and formulating three-dimensional tensors. The tensors are formed by stacking representations, such as layer activations, generated from several inputs and model 'snapshots', throughout the learning process. By conducting tensor component analysis (TCA), we aim to uncover meaningful patterns about how the internal representations evolve, expecting to highlight the merits or shortcomings of examined CL strategies. We plan to conduct our analyses across different model architectures and importance-based continual learning strategies, with a curated task selection, allowing us to gain insight into whether any observed patterns are consistently replicable.

## 1    INTRODUCTION

Learning is a core capability for all intelligent agents. While biological agents acquire new knowledge and adapt by building upon their prior learning experiences, artificial

agents follow a more static and meticulously curated learning process. When challenged to encounter new concepts, a setting familiar to biological agents, machine learning models suffer from *catastrophic forgetting*, demonstrating poor performance when attempting to recall old patterns (Schlimmer and Fisher, 1986; Ring, 1997). As continual learning (CL) research matures in tackling this challenge, solutions will look increasingly composite, likely layering multiple mechanisms to mitigate forgetting and achieve additional goals (Kudithipudi et al., 2022). The growing complexity underscores the need for an architecture and strategy agnostic explainability tool designed to shed light on how CL methods enable models to acquire new tasks while preventing the forgetting of previously learned ones. The insights gained from studying how representations change when experiencing catastrophic forgetting has previously facilitated the advancement of new methods. Thus, better understanding how existing methods update model parameters over time holds the potential to inspire the development of even more effective learning strategies.

To execute our study, we propose a novel framework that allows us to explore internal state changes during the continual learning process. Our framework centers on creating a data representation that captures the model's internal representations across different tasks over time. By leveraging Tensor Component Analysis (TCA) (Williams et al., 2018), a technique for three-dimensional tensor decomposition, we expect to uncover patterns concerning internal representations across time. The proposed methodology, as depicted in Figure 1, provides an overview of our study in understanding how representations evolve during CL.

**In summary, the contributions and the key insights that can be derived from this analysis are:**

- We leverage TCA, an unsupervised method for analyzing three-dimensional tensors, to extract patterns about how model representations evolve in a continual learning setting. TCA results in a data representation that potentially captures the model's

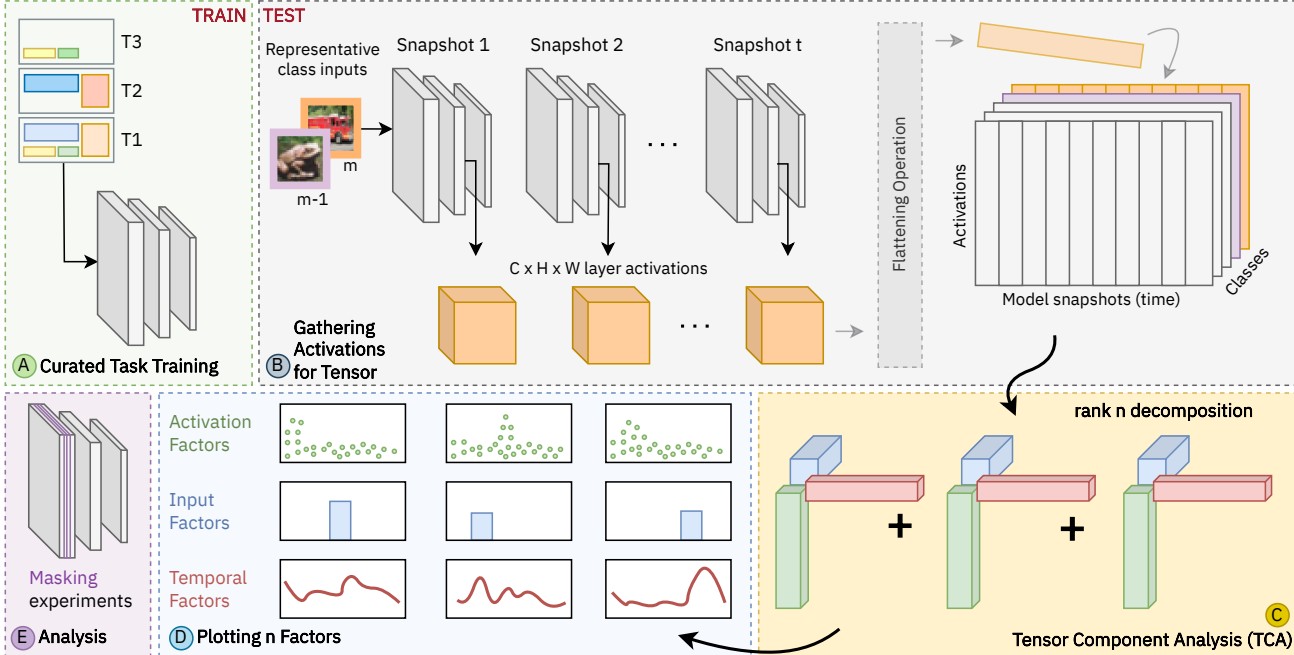

Figure 1: Overview of the proposed methodology. (A) We train a model with a CL method with curated tasks. (B) We assume access to 'snapshots' of the model taken throughout training on all tasks. At inference time, for a given dataset class, we feed its corresponding representative inputs into each of the snapshots and, for a layer of choice, gather the resulting activation tensors. We flatten the activations into a vector and stack them for all snapshots to obtain a matrix. We repeat this process for all inputs and stack the matrices to obtain a tensor. (C) We conduct tensor component analysis (TCA) with rank $n$ to obtain $n$ components. (D) We plot $n$ components, each of which consists of three factors: factors selecting for certain activations (green), factors selecting for certain inputs (blue), and factors describing a temporal activity (red). (E) We conduct model masking experiments to verify our observations.

dynamics for multiple inputs over time. To the best of our knowledge, we are the first to leverage TCA in continual learning interpretability.

- We systematically compare the performance, and analyze the internal representations of several parameter-based methods (and their combinations with replay) across convolutional neural networks (CNNs), vision transformers, and a CNN-transformer hybrid architecture. In addition, we conduct these experiments with a curated set of tasks to study how internal representations change based on the feature similarities of tasks.

- We investigate neuron tracking and filter evolution which can shed light on the neural plasticity that occurs during continual learning. We plan to gain insights on which neurons or filters are more flexible and likely to change their responses over time, and which ones remain stable.

**Summary of Hypotheses:**

**Hypothesis Set 1.** Do importance-based regularization methods lead to the emergence of 'specialized' neurons

for specific tasks? Does adding replay reinforce this specialization? Is there a consistent behavior across several model architectures? By tracking activations throughout training with various CL strategies, we will study the temporal patterns of these activations.

**Hypothesis Set 2.** Do CNN filters or transformer features within the same layer exhibit different update patterns throughout training? How do these update patterns compare by CL strategy and model architecture? By tracking what CNN filters and transformer features select, we will examine how filters/features evolve throughout the task-incremental learning.

## 2 BACKGROUND & RELATED WORK

### 2.1 Techniques for Examining Representation Quality

By examining representations, one may assess how well a model's learned features generalize across tasks and the degree to which they preserve task-specific information. Strong representations may enable better knowledge transfer and improved adaptation to new tasks. In this

**Nishant Suresh Aswani**[*,†]    **Amira Guesmi**[†]    **Muhammad Abdullah Hanif**[†]    **Muhammad Shafique**[*,†]

section, we explore several works studying representations in the context of CL. These studies employ various metrics and experimental scenarios to evaluate the quality of learned representations during training.

Ramasesh et al. (2020) studied the impact of catastrophic forgetting on hidden representations by quantitatively comparing the layer representations before and after sequentially training on a second task. To measure the similarity between representations across multiple layers of the model, they utilized the centered kernel alignment (CKA) technique. They also investigated how the semantic similarity between tasks affected forgetting, obtaining nuanced results suggesting that forgetting is maximized when sequential tasks have intermediate rather than low or high similarity.

Davari et al. (2022a) evaluated the quality of representations using Linear Probes (LP), which involves training a linear classifier on the fixed representations for each task. The accuracy of this linear classifier on the task's test set served as a metric to assess the representation quality for that particular task. To measure forgetting, the authors measured the difference in linear probe performance before and after introducing a new task.

Zhang et al. (2022) created a synthetic dataset in which alternating (odd and even) tasks shared the same low-level features while each task simultaneously contained unique, high-level features. By having access to the 'ground-truth' features, the authors examined whether the model was making progress towards learning the features encoded in the inputs. Their findings revealed that when the shared feature was consistently encountered in all even-numbered tasks, it prevented the model from fully learning the shared feature present in the odd-numbered tasks. The same held true for learning the shared feature in even-numbered tasks.

Hess et al. (2023) introduced a task exclusion comparison, hypothesizing that if a model has trained on a task and retained knowledge related to that task, then it should exhibit a more robust representation of that task compared to a model that has never encountered the task. To test this, they compared the linear probe accuracy of models that have encountered a task with those that have had the same task deliberately excluded. Their results indicate that continually trained models typically forget task specific knowledge quickly, contrary to what was presented by Davari et al. (2022a).

Across most works studying layer representations in the context of continual learning, CKA and linear probes make a recurring appearance as tools to measure the similarity of hidden layer activation patterns in neural networks. Although these metrics offer a broad understanding of representation comparisons, the occasional conflicting results among existing literature using similar experimental setups highlights the need for a different approach to studying CL dynamics. Recent representation similarity work has even highlighted the sensitivity of CKA to outliers (Davari et al., 2022b), urging researchers to ensemble similarity measurement using several techniques.

Nevertheless, most analyses of network representations in the context of CL limit themselves to similarity measurements of network representations pre and post-task training. We believe there is an opportunity to move away from drawing conclusions about network representations from ambiguous similarity measures when attempting to understand catastrophic forgetting. Instead, we suggest that it is possible to take advantage of unsupervised tensor decomposition to study how internal representations evolve across time for several inputs.

## 2.2 Tensor Component Anlysis (TCA) for Exploring Learning Dynamics

Tensor component analysis (TCA), also known as canonical polyadic (CP) decomposition, is a tensor decomposition technique to identify variability across the three axes of a tensor (Carroll and Chang, 1970; Harshman et al., 1970). While it is a dimensionality reduction technique similar to Principal Component Analysis (PCA), TCA differs by extending the decomposition to an additional axis. Further, unlike PCA, the factors obtained by TCA are not necessarily orthogonal, allowing for an expression of more natural patterns. For PCA, unless the features present in the data are naturally orthogonal to each other, the components recovered cannot be interpreted as those underlying features. In addition, PCA yields multiple solutions that can be employed to reconstruct the original data, known as the rotation problem (Williams et al., 2018). In practice, TCA does not face this limitation. While TCA is not entirely immune to alternative solutions, the number of potential solutions is more constrained, and in the case of non-negative TCA, the solution is typically unique (Kruskal, 1977; Adali et al., 2022). In various contexts within neural statistics, Williams et al. (2018) demonstrated that TCA serves as an unsupervised technique capable of demixing neural data and providing interpretable results corresponding to agent behavior and learning patterns.

While Williams et al. (2018) discussed TCA as a method to improve neural data analysis, such that it describes trial-to-trial variability and avoids a trial-averaging step, McGuire et al. (2022) took advantage of TCA in a slightly different fashion. Studying the behavior of neurons in the postrhinal cortex (POR) of mice, the authors attempted to identify neuron population clusters, as well as the clusters' responses over time to various cues displayed to the mice as they sequentially learned two tasks. Rather than focusing on trial-to-trial variability, the study focused on identifying within-stage and across-stage dynamics for multiple cues. In this context, 'stage' refers to the learning stages across

two tasks. Through TCA, the researchers observed a 'division of labor' in the neural populations, showing that certain clusters of neurons are specialized to activate in response to particular cues. Moreover, they were able to capture how these clusters vary their response across time.

Dyballa et al. (2023) used TCA as part of their analyses to organize neurons into a 'manifold' to study how the neurons they measured lie with respect to each other in a lower dimensional space. Generating these manifolds for biological and artificial neurons allowed the authors to compare the primary visual cortex (V1) to a CNN by studying the differences in how neurons are encoded within their manifolds. To accomplish this, the authors first employed TCA to obtain components which formed a lower dimensional space. Then, this component space was transformed into a manifold using the IAN similarity kernel and diffusion maps. The authors claimed that their analyses allowed them to organize neurons accounting for both stimulus features and temporal response, rather than the conventional approach of solely focusing on stimulus selectivity. In essence, the authors highlighted the ability of TCA to unlock an additional dimension, as well as provide outputs that can be fed into further analyses for network comparisons.

# 3 PROPOSED METHODOLOGY

Through tensor component analysis (TCA), we plan to investigate how the internal representations of CL models evolve as the models train on incoming tasks. As suggested earlier, TCA unlocks an opportunity to ask questions about learning patterns that might emerge in the temporal dimension while looking at several inputs. The main questions that we will explore in this investigation are:

- Are continual learning strategies able to elicit neuron specialization as they encounter new tasks?

- How do convolutional filters and transformer features shift over time when encountering new data in the continual learning setting?

For this exploration, we plan to use combinations of several CL strategies and model architectures. We look towards insights provided by simple explainability techniques to construct an understanding of continual learning dynamics.

## 3.1 Overview

Our proposed framework (see Figure 1) involves sequentially training a model on a stream of curated tasks (see Section 4) and analyzing its representations. To achieve this, we adopt the following steps:

**Saving Model Snapshots** During the continual training process, we periodically save snapshots of the model at defined intervals. For all tasks, we save multiple snapshots of the model while it is still learning the task, in order to conduct within-task and across-task analyses.

**Probing Model Snapshots** Then, we probe each model snapshot. This process involves feeding data to the model and recording, for example, the activations from a designated layer within the model. Activations represent the outputs of a layer in response to the given inputs. The neural activations collected during this process are structured into a three-dimensional tensor. As we will later elaborate, our analysis extends beyond neural activations.

**Tensor Component Analysis** Through tensor decomposition, we hypothesize deriving interpretable components. These components can, for instance, provide descriptions of how specific neural activations evolve throughout the learning process. Such analysis holds the potential to offer valuable insights into how changes in activations over time might correspond to the model's performance.

## 3.2 Problem Setup

We begin with the standard setup of a CL problem, as described by Hess et al. (2023), where we assume an incoming stream of classification tasks $\mathcal{T} = \{\mathcal{T}_1, \mathcal{T}_2, ..., \mathcal{T}_i, ..., \mathcal{T}_t\}$. Each task consists of a set of images $X_i$ with their corresponding class labels $Y_i$. In the case of a multi-head classifier, the model also has access to the unique task identifier $i$. Section 4 discusses how we curate our tasks to aid the analysis. For a chosen CL strategy, we train a model $f_\theta$ sequentially on these tasks. For the purposes of a temporal analysis, we save a snapshot of the model parameters $\theta_{i,e}$ at defined intervals, where $i, e$ refers to the parameters of a model learning a task $\mathcal{T}_i$ after having completed training epoch $e$. For instance, if we train a model for 50 epochs on each of 3 tasks and take snapshots at a frequency of 10 epochs, we would obtain a set of snapshots $\theta = \{\theta_{1,10}, \theta_{1,20}, ..., \theta_{3,40}, \theta_{3,50}\}$.

## 3.3 Tensor Formulation Details

One way to probe a model snapshot $\theta_{i,e}$ is with an input $m$ to obtain network activations $A_{i,e}^m$ from a desired model layer. We can repeatedly probe that layer with $n$ different inputs, to obtain $n$ different activations $A_{i,e} = \{A_{i,e}^1, A_{i,e}^2, ..., A_{i,e}^n\}$. If we flatten the neural activations, we will obtain a two-dimensional matrix of neural activations for multiple inputs. As shown in Figure 1, by repeating this procedure for all available model snapshots, we can build a three-dimensional tensor containing the neural activations across time for several inputs. Along one dimension of the tensor, one would vary the model snapshot $\{i, e\}$, and along another one would vary the input $m$. To gather neural activations for this analysis, it is essential to feed meaningful input into

**Nishant Suresh Aswani**[*,†]  **Amira Guesmi**[†]  **Muhammad Abdullah Hanif**[†]  **Muhammad Shafique**[*,†]

Table 1: **A Selection of Strategies Based on Parameter Importance**

| Approach | Strategy | + ER? | Ref. |
|---|---|---|---|
| Baselines | Naive | ✗ | |
| | Cumulative | ✗ | |
| Replay | Experience Replay (ER) | ✗ | |
| Importance-Based Regularization | Memory Aware Synapses (MAS) | ✓ | Aljundi et al. (2018) |
| | Elastic Weight Consolidation (EWC) | ✓ | Kirkpatrick et al. (2017) |
| | Synaptic Intelligence (SI) | ✓ | Zenke et al. (2017) |
| | Adaptive Group Sparse Regularization (AGS-CL) | ✓ | Jung et al. (2020) |
| Importance-Based Subnetworking | Relevance Mapping Networks (RMN) | ✓ | Kaushik et al. (2021) |
| | Winning Subnetworks (WS) | ✓ | Kang et al. (2022) |

the model. Future sections discuss potential strategies for selecting inputs. However, we emphasize that our analysis extends beyond neural activations.

## 3.4  Tensor Component Analysis

TCA approximates a three-way tensor $X$ as a sum of rank-1 tensors, where each rank-1 tensor is an outer product of vectors. The three-way tensor can be expressed as follows (Williams et al., 2018):

$$\hat{X} \approx \sum_{r=1}^{R} (\mathbf{u^r} \otimes \mathbf{v^r} \otimes \mathbf{w^r}) \tag{1}$$

In the formulation above, the vector $\mathbf{u^r}$ might represent patterns in neural activations, $\mathbf{v^r}$ represents the inputs corresponding to this activity, and $\mathbf{w^r}$ represents the stages of task learning where this activity is present. The rank $R$, a hyperparameter for this technique, determines the total number of components that approximate the original tensor.

The optimization objective for TCA aims to minimize the reconstruction error between the original tensor $X$ and its approximation $\hat{X}$, defined by the Frobenius norm:

$$\text{minimize} \quad ||X - \hat{X}||_F^2 \tag{2}$$

Additionally, the objective may be subject to non-negativity constraints on the component vectors $\mathbf{u^r}$, $\mathbf{v^r}$, and $\mathbf{w^r}$, i.e., all elements in these vectors are non-negative.

We will select a low rank $R$ with reasonable reconstruction error, as described in Section 4.

## 3.5  Activation Tracking

*Hypothesis 1: Does tracking neural activations throughout a continual learning setting, which constrains parameter changes based on an importance measure, reveal specialized classes of neurons?*

Importance-based methods for CL, such as Elastic Weight Consolidation (EWC) (Kirkpatrick et al., 2017), measure the importance of each parameter for a particular task and discourage certain parameters from significant changes throughout training or find meaningful masks for specific tasks. In our experiments, we investigate whether these strategies ultimately result in the emergence of sets of neurons that exhibit specialization for particular tasks. To explore this, we build a tensor with dimensions: [flattened neural activations] x [representative input(s) corresponding to dataset classes] x [model snapshot (time)]. We propose to conduct TCA on this tensor, thus expecting to capture patterns in neural activation of all the selected classes in a dataset across time.

In order to select the representative input(s) per class for which we will capture neural activations, we will experiment with the following approaches:

**Random Sampling** We will randomly select 20 images per class from the test dataset as the class representative inputs.

**Maximally Activating Example** We will search the test dataset for an image that maximally activates the desired class in the final model snapshot associated with the task that includes the class. For instance, if class 8 is a part of task $T_t$ trained over $e$ epochs, we would seek the test image that maximally activates the probability of class 8 in the model snapshot $\theta_{t,e}$.

**Class Optimized Image** We will optimize an image for

a fixed number of iterations, such that the resulting image maximizes the probability of a particular class, as described by Olah et al. (2017) for convolutional neural networks (CNNs).

When examining a TCA component, we hypothesize that specialization would be identified as a cluster of neurons characterized by greater activations. These activations would be associated with a segment of the temporal factors and one or few distinct inputs. This would indicate that certain neurons exhibited heightened activity during particular learning stages when the model was exposed to specific inputs.

### 3.6 Filter Evolution

*Hypothesis 2: Do CNN filters or transformer features within the same layer demonstrate equal levels of activity in their evolution throughout the continual learning process?*

In this experiment, we wish to understand how individual CNN filters and transformer features for a chosen layer evolve across the training regime. We ask if certain filters/features update earlier in the training regime and others update later or if filters/features are constantly changing. Specifically, do importance-based methods induce a markedly distinct updating pattern when compared to a purely replay-based approach or a baseline approach? Here, we build our tensor using dimensions: [flattened optimized image] x [filters/features] x [model snapshot (time)].

In contrast to the previous tensor formulation, this analysis will not be conducted for any specific class. Instead of examining multiple classes, we will focus on studying all the filters or features within a selected layer of the model. Additionally, rather than relying on neural activations, our approach involves utilizing optimized images specifically designed for each filter (in the case of CNNs) or feature (in the case of transformers) within a given layer. This optimization process is akin to the method of generating images that maximally activate a target class. In our case, we optimize images to achieve maximum activation for a chosen filter or feature within a model layer. This optimization procedure is inspired by the work of Olah et al. (2017) for CNN architectures and Ghiasi et al. (2022) for transformer architectures.

In this experiment, we may observe a scenario where a specific component exhibits the following characteristics: it selects one particular filter in the filter factors, corresponds to a particular segment in the temporal factors, and encompasses a subset of pixels within the 'flattened image' factors. Such a result could imply that this specific filter became actively engaged after learning a certain task. The temporal factors would indicate when it became active, the filter factors would specify which filter was active, and

the flattened image factors would reveal which region was active during this process.

## 4 EXPERIMENTAL PROTOCOL

**Datasets** We propose to run our experiments on the following classification datasets: (i) SplitMNIST, (ii) SplitCIFAR10, (iii) SplitCIFAR100, and (iv) twenty CIFAR100 superclasses (Ramasesh et al., 2020). The proposed settings cover a variety of task complexities. Across all comparable experiments, the dataset splits and task orders will be consistent to ensure a fair comparison.

**Task Generation and Order** Influenced by the approach of Ramasesh et al. (2020) and Zhang et al. (2022), we wish to apply our framework in a controlled setting by curating the order of tasks. Since we are interested in studying how learned features evolve, we propose that the initial task be large enough to learn rich features. We hypothesize that smaller initial tasks lead to poor initial model representations, making it difficult to draw meaningful conclusions. For instance, using Split-CIFAR10, the initial task may consist of the following four classes: airplane, automobile, cat, and horse, allowing the model to 'generalize.' We can then curate the next task to either be a two-way classification between deer/dog (animal) or truck/ship (vehicle). We can conduct a similar task curation for SplitMNIST and SplitCIFAR. In order to meaningfully curate these tasks, we propose generating t-SNE embeddings of the datasets, providing us with a coarse understanding of which classes might share features. Then, for instance, we can select dissimilar classes to consist the initial task and experiment with the remaining classes for the next tasks.

**Strategies** To study how various CL strategies affect model weights, this work will focus on the strategies outlined in Table 1. The selected strategies all exploit parameter importance to improve incremental learning, likely being an interesting class of strategies to study for our hypotheses. The final column describes whether there will be an additional variant of the strategy where it is combined with experience replay. We expect to finalize precise hyperparameters through a rigorous grid search.

**Model Architectures** We aim to study three architectures, selecting variants with a similar number of parameters: (i) ResNet-50 (23M parameters) (He et al., 2016), (ii) ViT-S/16 (22M parameters) (Dosovitskiy et al., 2020), and (iii) Convolutional vision Transformer (CvT)-13 (20M parameters) (Wu et al., 2021). For the experiments outlined in Sections 3.5 and 3.6, we will use multi-head models and provide task identities during the inference phase. We will train all models with the Adam optimizer using a batch size of 128 images

**Nishant Suresh Aswani**[*,†]   **Amira Guesmi**[†]   **Muhammad Abdullah Hanif**[†]   **Muhammad Shafique**[*,†]

and a fixed learning rate, which will be determined for each architecture by tracking the most accurate model on the validation sets of the scenarios. We expect to train models for a varying number of epochs depending on the architecture and the dataset (e.g. 30 epochs for SplitMNIST, 100 epochs for SplitCIFAR-100 with ResNet-50). However, we will ensure model training parameters (e.g. learning rate, number of epochs) are constant across strategies when comparing experimental results. Hyperparameters will be chosen such that the average task accuracies are within a comparable range (e.g. within 3%). We will equivalently seed models to ensure they are initialized with the same weights.

**Hyperparameter Selection Experiments** We plan to conduct extensive grid search experiments for hyperparameter selection, similar to those conducted by van de Ven et al. (2022), to ensure that the selected hyperparameters for the main experiments result in models with comparable performance across tasks. The primary focus of this work is to explore how internal representations are affected by the choice of continual learning strategy. Therefore, we will minimize explorations on the effects of various hyperparameters, instead aiming to pick a fair set of hyperparameters that will ensure reasonable and comparable model performance for a given architecture and dataset across continual learning strategies.

**TCA Model Optimization** We will explore fitting our TCA models with the nonnegative hierarchical alternating least squares (HALS) (Cichocki et al., 2007) and nonnegative block coordinate descent (BCD) algorithms (Lee and Seung, 2000), with the expectation to select the optimization algorithm that obtains the lowest minima solution for each experiment. We will remain consistent on our selection for a given architecture and dataset. We expect the nonnegative variant of TCA to provide the most interpretable results.

**TCA Rank Selection** Adopting from how McGuire et al. (2022) utilized TCA, for each experiment, we plan to fit TCA models across a range of ranks (e.g., 1-20) and plot the errors. We will empirically determine a narrower range of ranks where the "elbow" in the plot lies, essentially looking for the lowest number of ranks that provide reasonable error. For each rank within the determined "elbow-range", we will fit 10 TCA models and compare the similarities between the components returned. For the final results, we will select the lowest rank that consistently returns a high similarity (above 0.8) (Williams et al., 2018).

**Evaluation and Metrics** For all experiments, we plan to report the average of the final classification accuracy and backward transfer, as defined by Kang et al. (2022). We recognize that our framework is skewed towards an

empirical analysis, as we expect to extract activation and filter update patterns from plotting the resulting components after a tensor decomposition. Given that our hypotheses seek some level of specialization within the model, we propose to run masking experiments and measure the change in final accuracy to validate our findings from the TCA models. To elaborate, we can randomly mask certain filters in a CNN layer and measure the change in output accuracy. If our analysis reveals that specific filters hold significance for a particular class, the subsequent masking experiment could quantify the practical usefulness of our observation.

To compare the components from two TCA models, we will utilize the similarity score proposed in (Williams et al., 2018), which first solves an assignment problem to match components between two models and finds the 'optimal' permutation of components. Then, it computes the dot product between the matched components and returns the mean as a similarity score.

## 5 CONCLUSION

Our deliberate choice to work with importance-based strategies stems from their explicit goal to encourage specialization, such that certain model parameters establish greater importance for certain tasks. Our analysis is centered on the determining whether we are able to track this type of specialization to mechanistically understand if importance-based methods operate in the intended way. We believe specialization in CL is crucial because it encourages an efficient use of predefined resources as the model learns to allocate parameters to accommodate new tasks. By identifying meaningful behavior of existing methods through this analysis, we wish to pave the way to improve importance-based methods or discover the need to seek a different angle of attack to elicit specialization. Our work aims to find effective strategies for CL and resource allocation in machine learning models.

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
