# OpenReview forum: "Examining Changes in Internal Representations of Continual Learning Models Through Tensor Decomposition"
_continualai.org/CLAI/2023/Unconference_Preregistration_Track — 1st CLAI Unconf_

### Official Review · Reviewer_2PtH · 2023-08-15
**Unspecific hypotheses and unclear relevance.**

**Clarity:** 3
**Originality:** 3
**Soundness:** 1
**Significance:** 2
**Rating:** 4
**Confidence:** 4

**Review:**

The authors propose to investigate representations learned in continually trained neural networks and how they evolve during training. Their aim is to learn and understand how different continual learning algorithms influence the change in the activation values in the trained networks. Their main research questions are (1) are their 'specialized' neurons for specific tasks in neural networks and (2) how these filters or features change during training. Some existing techniques to study representations of neural networks in continual learning, such as CKA and linear probes, are discussed, among with relevant past work. The authors deem these tools insufficient and propose to use tensor component analysis as better analysis tool, which could be seen as a 3D extension of a PCA analysis, adding a temporal dimension.


It's interesting to see how tools that are used in biology to study neurons could also help explain artificial neural networks. The authors refer to two biology papers that have used this technology, but this seems unused in the setting of ANNs. The explanation in the paper on how to use TCA on neural network activations is relatively short, and I expected it to be longer given the lack of prior work. Especially since the time-axis of activations in biological neurons is not static, while those of artificial ones are, see Figure 4e in McGuire et al. (2022). So it's unclear to me how the 'time' and 'sample' axes need to be interpreted. Further, I miss a discussion on how the authors would interpret the results and a clear hypothesis of what the results would look like. I'll mention two examples for both research questions below.

RQ1, concerning specialized classes of neurons: there is no description, definition, or suggestion of what these specific neurons would look like. To answer 'Does X exist?', you need to precisely define what 'X' is, how else could you look for it? There's no hypothesis of what these would look like, which leaves a lot of room for any pattern to be recognized as a pattern that indicates specialization, even though it can also be something entirely different.

RQ2, concerning the 'activation' of filters or features. It is unclear to me how you would look for 'active' filters or features. In which of the three dimension of the TCA would I have to look to see which features are active and which aren't?. If I understand correctly, the authors would use an image that is optimized for a specific filter/feature in a snapshot of a model. So at that point in time, the filter is presumably going to be very 'active', because the used image is optimized precisely for that. But does that exclude possibilities where a filter in a CNN had changing weights during training, and thus it could be 'active' for another pattern, that is not in the image that is optimized and used in the analysis?

Finally, I miss a discussion on why this technique and these questions are going to be helpful in understanding and improving continual learning. In short, why is this necessary and why are the questions asked the ones that need answers?

**Strengths:**

* The proposal is unique and proposes a unique way of looking at the representations learned by a neural network.
* The problem setting is well defined, and many important details are clearly defined (e.g. models, data, algorithms...)
* Understanding continual learning and it's dynamics is relevant, as it might contribute to realizing a working continual learning algorithm of which we understand its limitations and possibilities.
* Overall, the paper is well written and easily to read.

**Weaknesses:**

* The main research questions are not rigorously defined and open for interpretation after the results are obtained. See review above.
* The TCA technique is not necessarily adapt to the setting the authors want to use it in, and there is little discussion on how it will be used and interpreted, see review above.
* The relevance of the answers to the research questions isn't sufficiently clear to me.

**Questions:**

* How do you define specific features and how would you recognize them in a TCA decomposition?
* How do you define active features and why is an optimized image the best way of looking for them?
* What is the relevance of these questions for continual learning as a whole?

**Protocol:**

The hyperparameters and the experimental settings are fine, but the research questions are not. They are proposed in a way such that it is not clear what would be a negative or positive result. They are very open, which leaves a lot of room for free interpretation of the results. This could lead to false interpretations of patterns in the results. See review above for a lengthier discussion.

---

### Official Review · Reviewer_HGmy · 2023-08-20
**Interesting angle to study representational forgetting.**

**Clarity:** 4
**Originality:** 3
**Soundness:** 3
**Significance:** 4
**Rating:** 7
**Confidence:** 4

**Review:**

## Summary

This paper tries to explore the questions about representational change in continual learning models. Previous methods explored this in a way using Linear probing and CKA. In this paper, the authors use Tensor decompositions by stacking activations and Features for different representative samples and study how it changes with time. A 3 dimensional principal components extracted and plotted to qualitatively view the patterns in representational change. For quantitative analysis they use masking to see the change in accuracy.

## Main hypothesis
Mainly this paper focuses on finding if there are any specialized neurons for a specific task. Secondly finding how CNN featres and transformer features change for different continual learning methods

**Strengths:**

* A different angle is provided to study the representational drift in continual learning models. Authors suggest that they are the ones firstly using Tensor decomposition to study catastrophic forgetting
* The provided experimental setup is well presented. Important architectures and regularization-based methods are presented to be studied in the experimental stage

**Weaknesses:**

* The selection of multi-head models in experiments is not well supported. It would be better if single-head and multi-head both compared to see how this affects representational change.
* The authors claimed that previous methods used final outputs to compare the representational forgetting. In addition to masking-based analysis of final accuracy, it would be better if authors find another way to calculate the quantitative similarity of extracted features.
* Unnessary space in the last line of introduction (understan ding)
* Can do these experiments on pre-trained models and see whether representation changes or not.

**Questions:**

What is the reason for using multi-head models?

**Protocol:**

The experimental protocol is well formulated. Adding experiments with large pre-trained models would be a great option to study the forgetting effect.

---

### Official Review · Reviewer_zFF6 · 2023-08-21
**Review of TCA framework**

**Clarity:** 2
**Originality:** 3
**Soundness:** 3
**Significance:** 3
**Rating:** 8
**Confidence:** 4

**Review:**

The idea is impressive and significant, as highlighted in the strengths below. However, there is room for improvement in terms of writing and presentation, as outlined in the weaknesses section. Additionally, more extensive experiments are needed.

**Strengths:**

- The overall idea is impressive and would provoke necessary future thinking about the internal state of continual learning models. This would encourage the community to better understand and interpret the current architecture in mitigating forgetting, thus inspiring future algorithms. Furthermore, continuously learning a robust feature extractor is one of the most important perspectives in CL, but is currently overlooked with the emergence of pre-trained model-based works, attribute learning works, and classifier learning works. This paper would encourage more efforts to be directed towards backbone learning. I would like to emphasize the significance of this idea.
- The proposed evaluated perspective, i.e. Activation Tracking and Filter Evolution, is valuable.

**Weaknesses:**

- The paper could engage general readers more effectively by providing more motivation and by establishing a stronger connection between this evaluation and the current literature in the first few paragraphs.
- It would be valuable to have a thorough review of the recent literature to explain answer the hypothesis.

**Questions:**

- What is the reason for choosing the ViT-S/16 and (CvT)-13 backbones? Would you also address the evaluation of algorithms specifically designed for these particular backbones?

**Protocol:**

- Since this is an evaluation paper, I would suggest including more baselines. Please refer to Table 1 in this paper for an example of how to summarize and present baselines.

Prabhu, Ameya, et al. "Computationally Budgeted Continual Learning: What Does Matter?." Proceedings of the IEEE/CVF Conference on Computer Vision and Pattern Recognition. 2023

---

### Decision · Program_Chairs · 2023-09-12

**Decision:**

Accept

**Comment:**

Dear authors,

Congratulations, your paper has been accepted at the ContinualAI Unconference 2023! We look forward to engaging in further discussions with you and others in the community.

Details will follow shortly regarding camera-ready versions. Please do take the feedback from reviews into account.

Thanks!